

# Using a holographic imager on a tethered balloon system for microphysical observations of boundary layer clouds

Fabiola Ramelli[1], Alexander Beck[1], Jan Henneberger[1], and Ulrike Lohmann[1]

[1]Institute for Atmospheric and Climate Science, ETH Zurich, 8092 Zurich, Switzerland

**Correspondence:** Fabiola Ramelli (fabiola.ramelli@env.ethz.ch)
Jan Henneberger (jan.henneberger@env.ethz.ch)

**Abstract.** Conventional techniques to measure boundary layer clouds such as research aircrafts are unable to sample in orographic or densely-populated areas. In this paper, we present a newly developed measurement platform on a tethered balloon system (HoloBalloon) to measure in situ vertical profiles of microphysical and meteorological cloud properties up to 1 kilometer above ground. The main component of the HoloBalloon platform is a holographic imager, which uses digital in-line holography to image cloud particles in a velocity independent sample volume, making it particularly well suited for measurements on a balloon. The unique combination of holography and balloon-borne measurements allows observations with high spatial resolution, covering cloud structures from the kilometer down to the millimeter scale.

We present observations of a supercooled low stratus cloud (high fog event) during a Bise situation over the Swiss Plateau in February 2018. In situ microphysical profiles up to 700 m altitude above the ground and at temperatures down to -8 °C and wind speeds up to 15 m s$^{-1}$ were performed. We were able to capture unique microphysical features from the kilometer down to the meter scale. For example, we observed cloud regions with decreased cloud droplet number concentration ($<0.5\,\overline{CDNC}$) and cloud droplet size at scales of 30-50 meters. These cloud inhomogeneities could arise from adiabatic compression and heating and subsequent droplet evaporation in descending air parcels. Moreover, we observed conditions favorable for the formation of boundary layer waves and Kelvin-Helmholtz instability at the cloud top. This potentially influenced the cloud structure on a scale of 10-30 kilometers, which is reflected in the variability of the CDNC.

## 1 Introduction

Boundary layer clouds play a key role in regulating the Earth's climate and controlling its weather systems and are important for many aspects of our daily life. First, low-level clouds are an important part of the Earth's radiation balance (Hartmann et al., 1992). For example, low stratus clouds cover an extensive area over ocean and land (Warren et al. 1986, Warren et al. 1988), can persist for several days (e.g. Bendix 2002) and cool the surface in the annual mean (e.g. Randall et al. 1984). Second, low visibilities associated with fog can impact road, ship and aviation traffic, causing accidents, delays or cancellations (e.g. Fabbian et al. 2007, Bartok et al. 2012). The resulting economic losses are comparable to those caused by winter storms (Gultepe et al., 2007). Moreover, with the constantly increasing contribution of photovoltaic power, reliable forecasts of low-level cloud cover are of increasing importance for the renewable energy sector (Köhler et al., 2017).





However, current state-of-the-art numerical weather prediction (NWP) models have major issues in predicting the exact time and location of the formation and dissipation of low-level boundary layer clouds (e.g. Bergot et al. 2007, Müller et al. 2010, Steeneveld et al. 2015, Román-Cascón et al. 2016). This is due to an incomplete understanding and a poor representation of the numerous processes occurring in boundary layer clouds, spanning from the microscale to the synoptic scale. The life cycle

of boundary layer clouds is a result of complex interactions among microphysical, thermodynamic, radiative, dynamic, aerosol and land surface processes. These processes are often not well parameterized in current operational NWP models, and the horizontal (Pagowski et al., 2004) and vertical (Tardif, 2007) resolution of these models is insufficient to cover the characteristic cloud scales. From an observational perspective, there is a need for additional comprehensive and high-quality observations of boundary layer clouds, especially of their vertical structure. Presently, most of the observations of boundary layer clouds

are performed by satellites. Satellites have a continuous spatial coverage and are useful to obtain climatologies of the optical and microphysical properties of clouds (Bendix 2002, Cermak and Bendix 2008). However, current satellite observations are typically too coarse to resolve scales below $250\,\mathrm{m}$ and have limitations in measuring cloud properties in the lowest kilometer of the planetary boundary layer (PBL) due to interference signals from the ground. Thus, in situ measurements of boundary layer clouds are important to gain a better understanding of the microphysical pathways in clouds.

Commonly, microphysical in situ measurements within the PBL are performed using a variety of measurement platforms, such as research aircraft (e.g. Sassen et al. 1999, Verlinde et al. 2007), helicopters (e.g. Siebert et al. 2006), cable cars (e.g. Beck et al. 2017), tethered balloon systems (TBS) (e.g. Siebert et al. 2003, Maletto et al. 2003, Lawson et al. 2011, Sikand et al. 2013, Canut et al. 2016) or launched balloon platforms (e.g. Creamean et al. 2018), each of which has its own advantages and disadvantages. For example, research aircraft can travel large distances and freely choose their flight path, but have minimum

altitude constraints, which limits observations within the lowest kilometer of the PBL. Moreover, due to high travelling speeds ($100\,\mathrm{m\,s^{-1}}$), aircraft measurements have limited spatial resolution and can be influenced by ice shattering on the instrument tips (Korolev et al., 2011). To investigate small-scale processes in clouds, measurement platforms with lower true air speed are advantageous. The aspiration speed on cable cars ($10\,\mathrm{m\,s^{-1}}$) is one order of magnitude lower than on aircraft, which enables probing the cloud with a much higher spatial resolution (Beck et al., 2017). However, the locations of cable cars are limited to

mountain areas. TBS can achieve a similar instrumental resolution as cable cars and are more flexible in terms of choosing the measurement location. Measurements with TBS can cover the full vertical extent of the PBL from the surface up to 1-2 kilometers. However, conventional, blimp-like TBS are limited to wind speeds below $10\,\mathrm{m\,s^{-1}}$ due to the instability of the balloon at higher wind speeds (e.g. Lawson et al. 2011, Canut et al. 2016, Mazzola et al. 2016). Moreover, TBS can be deployed further away from the ground, reducing the effects of surface-based processes such as blowing snow (Lloyd et al. 2015, Beck et al.

30 2018).

In this paper, we present a newly developed measurement platform for boundary layer clouds (HoloBalloon), consisting of a holographic cloud imager and a meteorological instrument package on a kytoon. Kytoons are a hybrid-balloon-kite combination allowing stable flight in wind speeds up to $30\,\mathrm{m\,s^{-1}}$. The stability in high wind speeds makes kytoons a promising measurement platform for cloud research, especially in locations with strong wind conditions (e.g. mountain regions). Due to

the low aspiration velocities of TBS, the choice of instrument is of particular importance, since fluctuations in wind speed and





direction could influence the measurements. Most cloud probes use an inlet to ensure a steady sampling velocity in fluctuating wind speeds (Baumgardner et al., 2011). However, the use of inlets increases measurement uncertainty, due to size-dependent particle losses at the inlet and non-isokinetic sampling effects. One technique that overcomes this problem is digital in-line holography, which provides a well-defined sample volume independent of particle size and aspiration velocity, making holo-
graphic cloud imagers particularly well suited for measurements on TBS. Digital in-line holography can simultaneously capture single particle information (position, size and shape) of an ensemble of cloud particles within a three-dimensional detection volume. Thus, it provides information of the phase-resolved cloud properties such as number concentration, size distribution, and water content (e.g. Beck et al. 2017), as well as the spatial distribution of cloud particles in a cloud volume on a millimeter scale (e.g. Beals et al. 2015). Digital holographic cloud imagers have been used in previous field campaigns on ground-based
(e.g. Thompson 1974, Kozikowska et al. 1984, Borrmann et al. 1993, Raupach et al. 2006, Henneberger et al. 2013, Schlenczek et al. 2017), airborne (e.g. Conway et al. 1982, Fugal and Shaw 2009, Beals et al. 2015, Glienke et al. 2017) and cable car (Beck et al., 2017) platforms, but have not yet been deployed on TBS.

The HoloBalloon platform merges the advantages of holography (well-defined sampling volume, spatial distribution) with the benefits of a TBS (high-resolution measurements) with the aim to observe the cloud structure on different scales. Information
about the macroscopic cloud structure can be obtained from the vertical profiles up to 1 kilometer above the ground and information about the cloud microstructure can be extracted from the analysis of the cloud particle spatial distribution within a single hologram on a millimeter scale. The HoloBalloon platform was tested in boundary layer clouds over the Swiss Plateau. Here we present observations of a case study during a stratus cloud (high fog) event. The cloud structure is analyzed on different scales, starting with the large-scale cloud structure of tens of kilometers and moving down to the cloud microstructure on the
meter scale. A particular emphasis is placed on cloud inhomogeneities. Previous observations found microphysical inhomogeneities on scales of a few tens of meters (e.g. Korolev and Mazin 1993, García-García et al. 2002, Gerber et al. 2005) or even on the sub-meter scale (e.g. Baker 1992, Brenguier 1993, Beals et al. 2015, Beck et al. 2017), which were attributed to different physical processes such as turbulent mixing or entrainment. These inhomogeneities can influence the cloud microphysics on different scales. For example, on a millimeter scale, they can be of importance for particle growth by collision-coalescence
and thus for the efficiency of precipitation formation. Inhomogeneities at scales of hundreds of meters and kilometers can be important for radiative heating and cooling. In this paper, we investigate whether similar cloud inhomogeneities are found in stratus clouds and aim to understand the formation of such inhomogeneities.

The first part of the paper introduces the HoloBalloon measurement platform (Sect. 2). The working principle and the setup of the newly developed holographic cloud imager is described in Sect. 3. Observations of a case study in stratus clouds during
a Bise situation obtained with HoloBalloon are presented in Sect. 4. These observations are discussed in a larger context in Sect. 5.





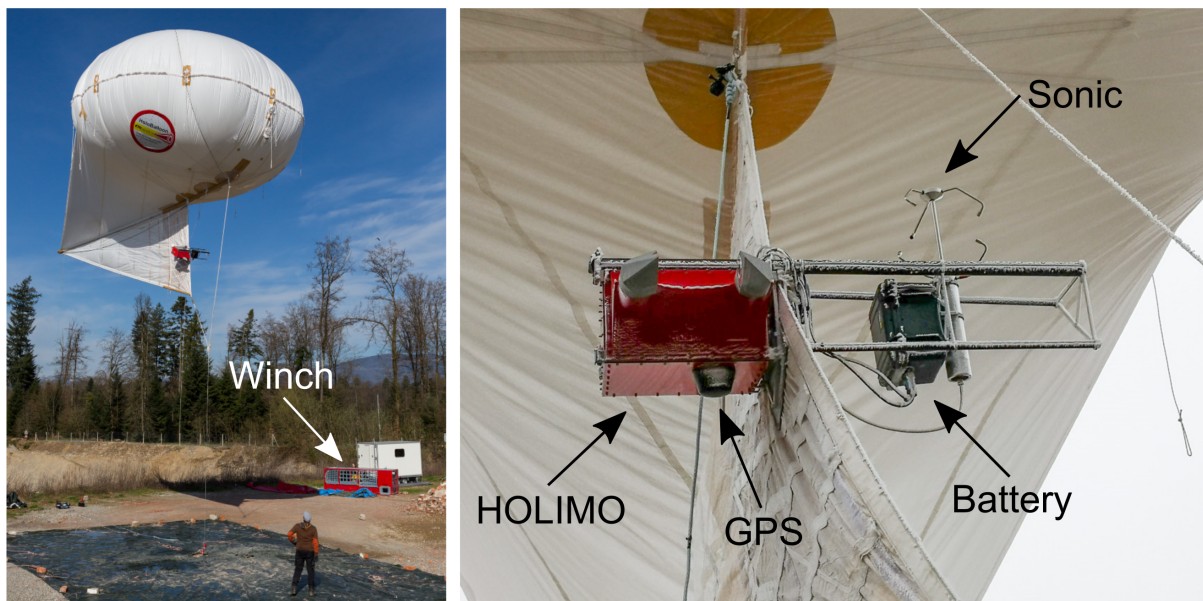

**Figure 1.** Experimental setup of the HoloBalloon platform consisting of a tethered balloon system (left) and the instrument package (right). The winch is visible in the left picture. The instrument package includes the holographic cloud imager HOLIMO 3B, a 3D sonic anemometer as well as a temperature and humidity sensor (not visible). The left picture has been taken by Pascal Halder (naturphotos.ch)

## 2 Description of the HoloBalloon measurement platform

The HoloBalloon platform is designed to obtain vertical, in situ profiles of the microphysical and meteorological cloud properties of boundary layer clouds up to 1 kilometer above ground. Our TBS consists of a 175 m³ kytoon (Desert Star, Allsopp Helikite, UK), a 1200 m long Dyneema cable and a gasoline winch to launch and recover the TBS (see Fig. 1). The balloon

has a net lift of 85 kg at sea level. Kytoons are a hybrid combination of a helium balloon and a kite, exploiting both for lift. The helium balloon creates static lift, while the kite creates aerodynamic lift in wind. The kite utilizes a long keel to provide stability in high wind conditions. The maximum operational wind speed of our TBS is 25 m s⁻¹. A further advantage of the kite is that it ensures that the instrument platform is oriented into the wind, allowing for the spatial distribution of cloud particles to be assessed.

The cable and winch are designed to withstand forces up to 4 tons, which can occur during high wind speed conditions (> 15 m s⁻¹). The 7 mm Dyneema line has a length of 1200 m and a breaking strength of 8200 kg. At wind speeds larger than 5 m s⁻¹, the TBS can have a flight angle of up to 45° due to the kytoon design, reducing the maximum flight height to 850 m. A system of three Platipus anchors is used to secure the balloon to the ground. The tethered balloon is launched and retrieved with a winch powered by a V8 Chevy engine (Skylaunch, UK). The winch has a line speed of 1 m s⁻¹ forward and reverse,

which allows a vertical profile of 500 m in 8 minutes.

The instrument package is installed at the kiel of the HoloBalloon platform. The key component is the HOLographic Imager



for Microscopic Objects (HOLIMO 3B) (see Sect. 3) which can measure phase-resolved cloud properties. Additionally, the HoloBalloon platform is equipped with a meteorological instrument package (see Fig. 1) consisting of a 3D sonic anemometer (THIES, 4.3830.20.340) and a heated temperature and humidity sensor (HygroMet4, Rotronic) in an actively ventilated radiation shield (RS24T, Rotronic). The platform is powered by a 1000 Wh battery, which allows for continuous operation of the

instrument package for up to 5 hours. Data are temporally stored on a 4 TB solid state drive and a mobile router enables remote access of the platform via a mobile data network connection (similar to Beck et al. 2017). The HoloBalloon instrument platform has a total weight of about 22 kg, consisting of the HOLIMO 3B instrument (13 kg), the meteorological instrumentation (5 kg) and the battery pack (4 kg).

To obtain reliable measurements of wind speed and direction the motion of the balloon needs to be removed (e.g. Canut et al.

2016). Here we used a GPS antenna (TW3740, Tallysman) and an inertial navigation system (Ellipse2-N, SBG systems) to measure the position, velocity and orientation of the instrument package. The GPS antenna and the inertial navigation system are fixed on the HOLIMO instrument and are thus an integral part of the instrument package. We followed the procedure described in Elston et al. (2015) to convert the wind measurements from the inertial frame to the sonic anemometer frame and thus to correct for the motion of the balloon. The corrected wind measurements are presented and compared to other observations

in Sect. 4.2.

## 3   HOLographic Imager for Microscopic Objects

The main component of the HoloBalloon instrument package is the holographic cloud imager HOLIMO 3B which can image cloud particles between 6 µm and 2 mm within a three-dimensional detection volume. Despite its open path configuration, HOLIMO has a velocity independent sample volume. This property makes HOLIMO particularly well suited for application

on a TBS due to fluctuating aspiration speeds towards a TBS.

### 3.1   Working principle of holography

HOLIMO works on the principle of digital in-line holography (Fig. 2), which consists of a two-step process requiring a coherent light source and a digital camera. In the first step, the interference pattern of a reference wave (laser) and a scattered wave (the light scattered by a cloud particle in the sample volume) is recorded as a hologram. The second step involves a reconstruction

process in which the 2D shadowgraphs and 3D in-focus position of the particles are extracted from the interference pattern, using the HoloSuite software package (Fugal et al. 2009, Schlenczek 2018). The resulting 2D shadowgraphs can be classified as cloud droplets, ice crystals and artifacts using supervised machine learning (e.g. Fugal et al. 2009, Beck et al. 2017, Touloupas et al. 2019 (submitted). From that, we can calculate phase-resolved microphysical properties such as number concentration, water content and size distribution. Moreover, because holography provides a snapshot of an ensemble of cloud particles within

a volume, the spatial distribution of the cloud particles can be recovered from the interference pattern. Unlike light scattering instrumentation, no assumptions about the particle shape, orientation or refractive index are required, as an image of the cloud particles is captured. The major disadvanted of holography is the high computational power associated with the reconstruction





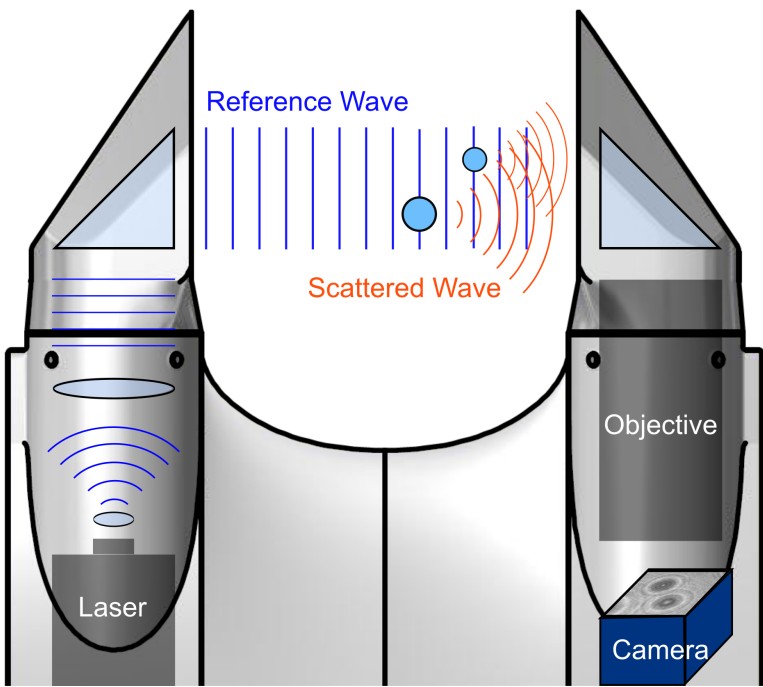

**Figure 2.** Schematic of the working principle of digital in-line holography. A collimated laser beam is scattered by two particles. The scattered waves interfere with the reference wave and form an interference pattern (i.e. a hologram) which is recorded by a digital camera.

process and the data analysis. The working principle of digital in-line holography and HOLIMO have been described in more detail in Fugal et al. (2009), Henneberger et al. (2013) and Beck et al. (2017).

### 3.2   Instrument description

A series of holographic instruments have been developed in the Atmospheric Physics group at ETH Zurich in the last decade
5  (Amsler et al. 2009, Henneberger et al. 2013, Beck et al. 2017). HOLIMO consists of two main units: The control unit, which comprises the temperature control system and the control and data-acquisition computer, and the optical imaging unit, which is integrated in the two instrument towers. As the previous version (HOLIMO 3G, Beck et al. 2017), HOLIMO 3B has an open path configuration. In contrast to the previous versions, HOLIMO 3B uses a 355 nm laser and an improved optical system to enlarge the detection volume and improve the optical resolution of the instrument.
10  A schematic of the optical system of HOLIMO 3B is shown in Fig. 2. The laser (FTSS355-Q4_1k, CryLas, Germany) emits pulses with a wavelength of 355 nm, with a pulse width of 1.4 ns and a pulse energy of 42 μJ. The beam is attenuated by a neutral density filter and focused through a 10 μm diamond pinhole (LenoxLaser HP-3/8-DISC-DIM-10), which acts as a point light source. The diverging laser beam is expanded by a biconcave lens and collimated to a beam diameter of 40 mm. After passing through a turning prism and a sapphire window, the collimated laser beam traverses the sample volume, before



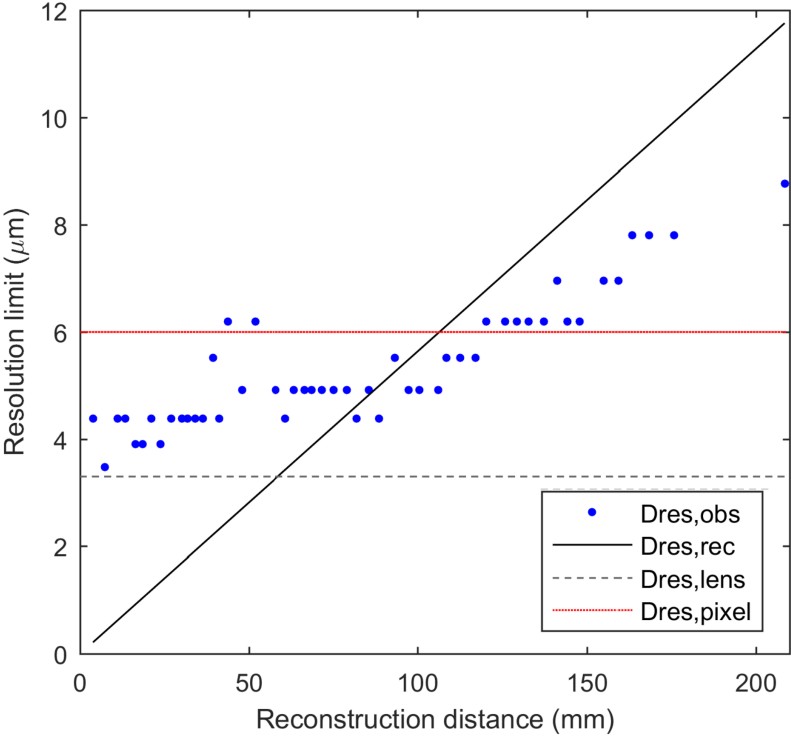

**Figure 3.** Optical resolution measurements of the HOLIMO 3B instrument as a function of the reconstruction distance. The blue dots represent the resolutions measured with a US Air Force resolution target 1951 USAF ($D_{res,obs}$). The three lines indicate theoretical resolution limits due to the pixel size ($D_{res,pixel}$, red solid line), the optical limitation of the lens system ($D_{res,lens}$, grey dashed line) and the optical setup of the instrument ($D_{res,rec}$, black solid line). The strongest resolution limit constraint determines the optical resolution of the instrument at a specific reconstruction distance. More information about the theoretical resolution constraints for holographic systems can be found in Henneberger et al. (2013) and Beck et al. (2017).

entering the imaging lens system in the opposite tower of the instrument. The bi-telecentric lens system (Correctal S5LPJ2755, TDL65/1.5 UV, Sill Optics, Germany) has a magnification of 1.5 and a numerical aperture of 0.13. The holograms are recorded with a 25 MP camera (hr25000MCX, SVS-Vistek, Germany) with $5120 \times 5120$ pixels, a pixel pitch of 4.5 µm and a maximum frame rate of 80 fps. The quadratic cross-sectional area of the camera allows for more uniform illumination of the edges than a

5   rectangular camera image, which was used in the previous versions.

The optical resolution of the system was tested using a US Air Force resolution target (1951 USAF), which is placed at different positions inside the detection volume, following the procedure described in Spuler and Fugal (2011) and Beck et al. (2017). The optical system described achieves a resolution ($D_{res,obs}$) of 6 µm within the first 110 mm of the reconstruction distance (see Fig. 3). This is consistent with the theoretical resolution limit of the pixel size ($D_{res,pixel}$). For reconstruction

10   distances larger than 110 mm, the resolution limit decreases and is determined by the resolution limit from the diffraction





aspects of in-line holography ($D_{res,rec}$). In general, the measured optical resolutions are in good agreement with the theoretical resolution constraints. Particles within the first 10 mm and close to the image border (< 0.2 mm from image edges) are not included in the analysis due to flow distortion effects from the towers and edge effects. With an effective cross-sectional area of 15 mm × 15 mm and an effective depth of 100 mm, this results in a sample volume of 22.5 cm$^3$ and a maximum sample

volume rate of 1800 cm$^3$s$^{-1}$ (with 80 fps).

### 3.3 Size calibration of HOLIMO 3B

Accurate sizing of cloud particles is important to obtain reliable measurements of cloud properties such as water content and size distributions. For holographic instruments, the sizing algorithm should be precise and accurate over a large particle size range (6 μm - 1 cm) and applicable for the entire detection volume. The sizing of the particles strongly depends on an amplitude

threshold value that separates particle pixels from background pixels. From the number of particle pixels, the area-equivalent diameter is derived. In the standard HoloSuite version, a uniform amplitude threshold is used for particle detection and particle sizing. However, a uniform amplitude threshold leads to unsatisfying results for particle sizing due to a decreasing signal-to-noise ratio with increasing reconstruction distance $z$ in the large detection volume of HOLIMO. This has the effect that the amplitude image of the particles becomes less distinct with larger $z$ distances and thus the observed particle size decreases with

increasing $z$ distance. To overcome this issue and to ensure a uniform sizing of the particles over the entire detection volume, Beck (2017) introduced a new method by normalizing the in-focus particle image. In the normalization step, the darkest particle pixel is set to 0 (black), the mean of the background pixels is set to 1 (white) and the rest of the pixels are scaled relatively. This results in a more uniform signal-to-noise ratio and allows applying a uniform amplitude threshold. The amplitude threshold can be used as a tuning parameter to calibrate the sizing algorithm of the HoloSuite software for the HOLIMO 3B instrument.

The sizing algorithm was calibrated using a Vibrating Orifice Aerosol Generator (VOAG model 3450, TSI, Minnesota, USA) for particle generation and an Aerodynamic Particle Sizer (APS model 3321, TSI, Minnesota, USA) for particle sizing. Particles with diameters between 5 μm and 18 μm were generated by the VOAG using a liquid oil-water solution. The generated particles were introduced into a 120 mm × 1000 mm cylindrical tube and measured by the HOLIMO 3B instrument and an APS that were installed at the end of the tube. The APS covers the size range between 1 μm and 20 μm and is used as a reference

measurement. Thus, the amplitude threshold was used as a tuning parameter to fit the HOLIMO measurements to the APS measurements. An amplitude threshold of 0.47 was found to fit the APS data best (smallest sum of squared errors).

The size distributions of the calibration experiments are shown in Fig. 4 and are summarized in Table 1. The size distributions of the HOLIMO and APS instruments were normalized to their maxima and a Gaussian distribution was fitted to the data. The results of the HOLIMO 3B instrument agree with the mean diameter of the APS within instrumental uncertainty. In

general, a trend towards an underestimation of the particle diameter compared to the APS is observed, except for the calibration measurements at the measurement limits of HOLIMO (6 μm) and the APS (18 μm). The overestimation of the particle diameter by HOLIMO for 6 μm particles may be due to the optical resolution limit of the HOLIMO 3B instrument. While HOLIMO 3B can only detect particles larger than 6 μm, the APS can detect particles down to a diameter of 1 μm. On the other hand, the overestimation of the particle diameter at 18 μm could be caused by a bias of the APS instrument, which has an upper





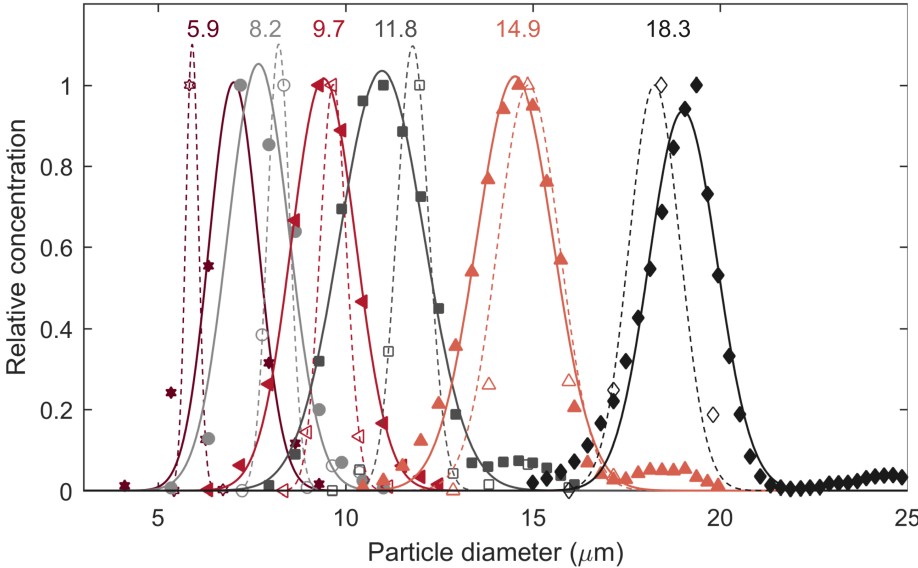

**Figure 4.** Size distributions from calibration experiments of the HOLIMO 3B instrument. The symbols show the normalized particle concentration measured by HOLIMO 3B (filled) and the APS (unfilled) instrument. The lines indicate the Gaussian distributions fitted to the HOLIMO data (solid) and APS data (dashed). The numbers represent the mean diameter of the APS size distribution.

detection limit of $20\,\mu m$. Thus, particles in the second peak at $23\,\mu m$ are not detected by the APS (see Fig. 4). To conclude, no correction to the sizing algorithm was made, because all size measurements agree within the square root of the pixel size ($\sqrt{3.01\,\mu m} = 1.73\,\mu m$).

## 4 Case study - Supercooled low stratus clouds

5 In this paper, we present observations of a supercooled low stratus cloud event (also referred to as high fog) during a Bise situation over the Swiss Plateau, obtained on 24 February 2018 between 08 and 10 UTC. The case study focuses on 9 vertical profiles of microphysical and meteorological cloud properties measured by the HoloBalloon platform. The analysis starts with

**Table 1.** Results of the size calibration experiments of HOLIMO 3B and an APS. The mean diameter and the standard deviation are derived from a Gaussian fit to the normalized size distribution.

| | Particle diameter (μm) | | | | | |
|---|---|---|---|---|---|---|
| HOLIMO 3B | $7.02 \pm 0.93$ | $7.68 \pm 1.15$ | $9.41 \pm 1.21$ | $10.97 \pm 1.59$ | $14.52 \pm 1.45$ | $19.01 \pm 1.28$ |
| APS | $5.91 \pm 0.24$ | $8.21 \pm 0.42$ | $9.67 \pm 0.50$ | $11.79 \pm 0.60$ | $14.88 \pm 1.18$ | $18.25 \pm 0.96$ |


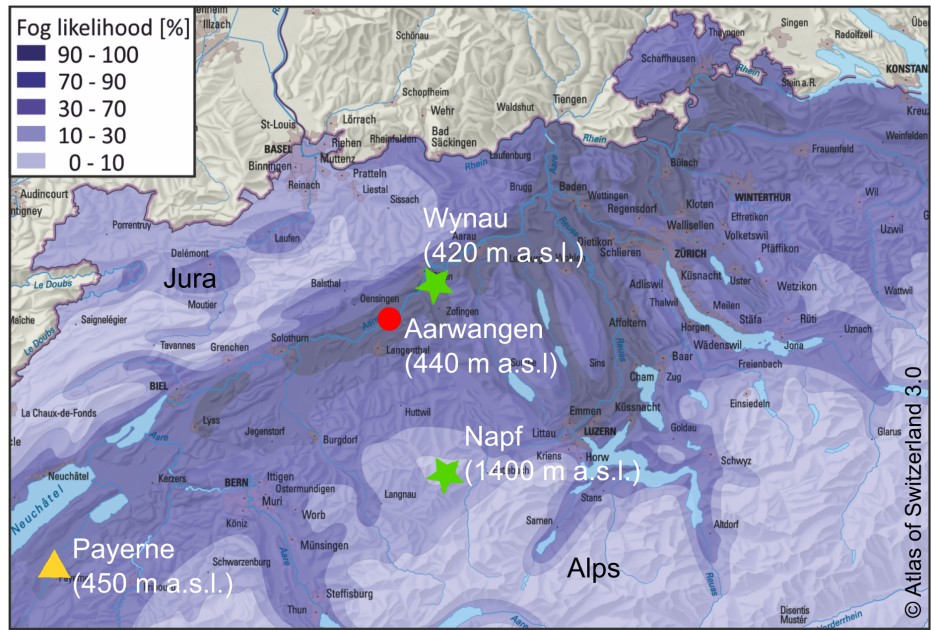

**Figure 5.** Map of the fog frequency during winter (adapted from the 'Atlas of Switzerland 3.0') and of the measurement locations. The climatology of fog is based on satellite images. The field locations comprise the measurement site in Aarwangen from which HoloBalloon was launched (red circle), ground-based weather stations from MeteoSwiss providing measurements of meteorological parameters (green stars) and the field site in Payerne from which radiosondes are launched twice a day (yellow triangle).

large scales, giving an overview of the synoptic weather situation and the large-scale cloud structure, and moves towards smaller scales, providing information about the cloud microstructure.

The Swiss Plateau, which lies between the Jura mountains and the Swiss Alps, is often covered by fog or low stratus clouds during autumn and winter due to its geographic location. A satellite-based climatology of fog and low stratus cloud coverage over the Swiss Plateau during high pressure situations in winter is shown in Fig. 5. In regions along rivers and lakes, a fog frequency of up to 90 % is observed. Most commonly, fog forms by radiative cooling during clear nights. Additionally, cold air flows from the alpine valleys and the Jura towards the Swiss Plateau, where the cold air can accumulate. This cooling of the air can cause condensation and the formation of ground fog. However, the case study presented here was connected to a Bise situation; a cold, dry east and north-east wind. During Bise, cold air is advected and pushed under warm air, leading to the formation of a strong temperature inversion. The cold air in the lower layer cannot easily escape the Swiss Plateau because it is bound by the Jura mountains and the Swiss Alps. If the air is sufficiently moist, condensation sets in and high fog or low stratus clouds can develop. The top of the cloud layer is defined by the height of the temperature inversion. The solar radiation reaching the boundary layer is often too weak to dissipate the fog layer in autumn and winter. Thus, ground fog or stratus clouds can persist for several days, until a change in the synoptic weather pattern occurs.





**Table 2.** Summary of the start and end time of the 9 vertical profiles taken with the HoloBalloon platform.

| Profile number | Profile type | Start time (UTC) | End time (UTC) |
|:---:|:---:|:---:|:---:|
| 1 | ascending | 08:01 | 08:10 |
| 2 | descending | 08:11 | 08:23 |
| 3 | ascending | 08:24 | 08:34 |
| 4 | descending | 08:35 | 08:45 |
| 5 | ascending | 08:46 | 08:58 |
| 6 | descending | 08:59 | 09:11 |
| 7 | ascending | 09:12 | 09:22 |
| 8 | descending | 09:23 | 09:37 |
| 9 | ascending | 09:38 | 09:57 |

## 4.1 Measurement location and data analysis

The measurements with the HoloBalloon platform were performed in Aarwangen (47°14' N, 7°45' E) in the Swiss Plateau 40 km northeast of Bern (Fig. 5). The field site is located at a gravel station next to the Aare river at an altitude of 440 m a.s.l. and is surrounded by grassland and forests. The balloon measurements were performed in a temporarily closed air space of
2 km in diameter, which was activated on measurement days. The maximum flight height allowed was 700 m above ground because of air traffic regulations. The experimental setup of the HoloBalloon platform is shown in Fig. 1.

The measurements taken on the HoloBalloon platform were complemented and validated by observations of surrounding MeteoSwiss weather stations and radiosondes (see Fig. 5). The weather stations are located within a radius of 30 km from Aarwangen and cover altitudes between 420 m a.s.l. and 1400 m a.s.l. to obtain an overview of the regional weather situation.
Radiosondes are launched twice a day (00 and 12 UTC) from Payerne, which is located 80 km south-west of Aarwangen. We used the radiosondes to determine the inversion height and thus the cloud top height, because we were not able to measure the whole cloud layer due to the air traffic restrictions on flight height.

A total of 9 microphysical profiles measured with the HoloBalloon platform was analyzed in this case study, with an average of 800 holograms (~5 L sampled volume) or 600'000 cloud particles per profile. Each profile had a duration of 10-15 minutes.
With a mean horizontal wind speed of $10\,\mathrm{m\,s^{-1}}$, this results in a horizontal resolution of around 6-9 kilometers. The start and end times of the individual profiles are summarized in Table 2. At least 10 holograms were grouped together to obtain better counting statistics. This results in a vertical resolution of 5 m. Only data points with a liquid water content (LWC) larger than $0.01\,\mathrm{g\,m^{-3}}$ (definition for cloud base) are considered in the analysis. Cloud particles smaller than 25 μm were classified using support vector machines, whereas particles larger than 25 μm were classified by hand. Only particles within a reconstruction
distance between 20 mm and 50 mm were included in the analysis. A smaller detection volume than described in Sect. 3.2 was chosen due to a mean droplet size close to the instrumental resolution limit and noise in the holograms.





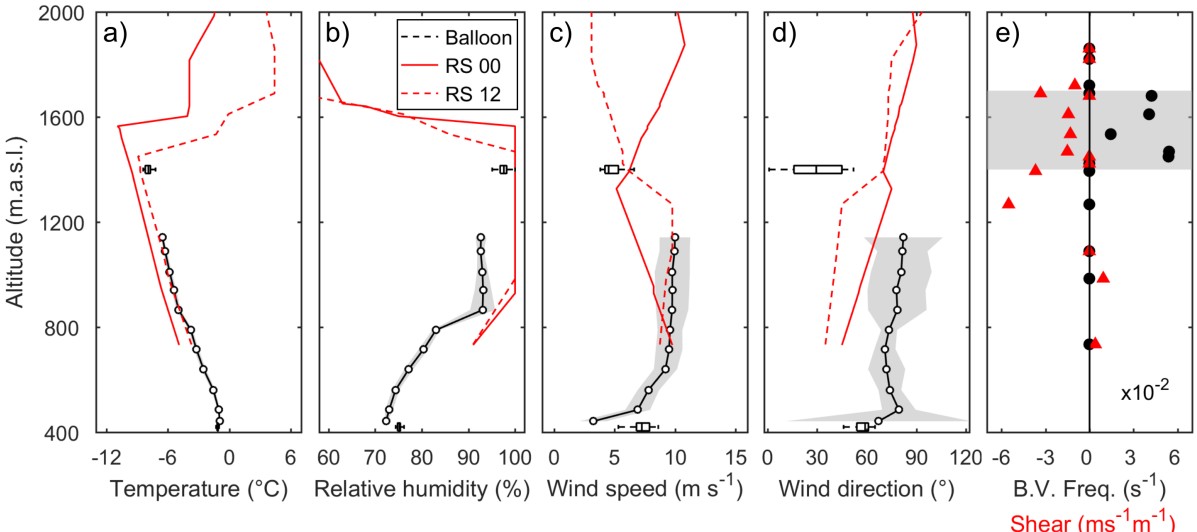

**Figure 6.** Vertical profiles of the meteorological parameters (a-d). The HoloBalloon measurements are averaged over 9 profiles and an altitude interval of 75 m. The black dots indicate the mean and the shaded area the standard deviation of the data. The vertical profiles of two radiosonde ascents (00 UTC (solid) and 12 UTC (dashed)) are shown by the red lines. The box plots represent the data from MeteoSwiss weather stations (Wynau (420 m a.s.l.), Napf (1400 m a.s.l.)). On each box, the central line indicates the median and the left and right edges of the box mark the 25th and 75th percentiles, respectively. The whiskers extend to the minimum and maximum of the data not considered as outliers. Figure 6e shows the vertical profile of the Brunt-Väisälä frequency ($N = \sqrt{\frac{g}{\theta}\frac{\delta\theta}{\delta z}}$) and the wind shear ($s = \frac{\delta v}{\delta z}$) calculated from the radiosonde ascent at 12 UTC. The shaded area indicates regions with a positive Brunt-Väisälä frequency.

## 4.2 Meteorological situation

Figure 6 shows the vertical profiles of the meteorological parameters during the measurement period. The meteorological conditions during the 2-hour measurement period were relatively stable. The temperature profile was characterized by a strong temperature inversion, which was located at around 1450 m a.s.l. The temperature varied between -1 °C at the surface and -8.9 °C at the inversion base. The height of the temperature inversion defines the top of the cloud layer. The relative humidity increased from the ground up to 850 m a.s.l., where it remained constant up to the inversion. We assumed that this constant relative humidity interval indicates conditions of water saturation and thus marks the extent of the cloud layer. No relative humidity values above 95 % were observed by the HoloBalloon platform. This can be explained by the challenges of measuring relative humidity at in-cloud conditions (e.g. Korolev and Mazin 2003, Korolev and Isaac 2006). Wind speeds between 6.7 m s$^{-1}$ and 8.6 m s$^{-1}$ were observed in Wynau with wind gusts up to 10.6 m s$^{-1}$. The wind speed in Aarwangen increased in the first 200 m above the ground from 7 m s$^{-1}$ to 10 m s$^{-1}$. As it can be seen from the radiosondes, the wind speed was relatively constant up to the inversion layer. The prevailing wind direction was north-east with a slight turn towards east with increasing altitude. At the inversion, a change in the horizontal wind speed and direction with height (vertical wind shear) occurs. In this region,

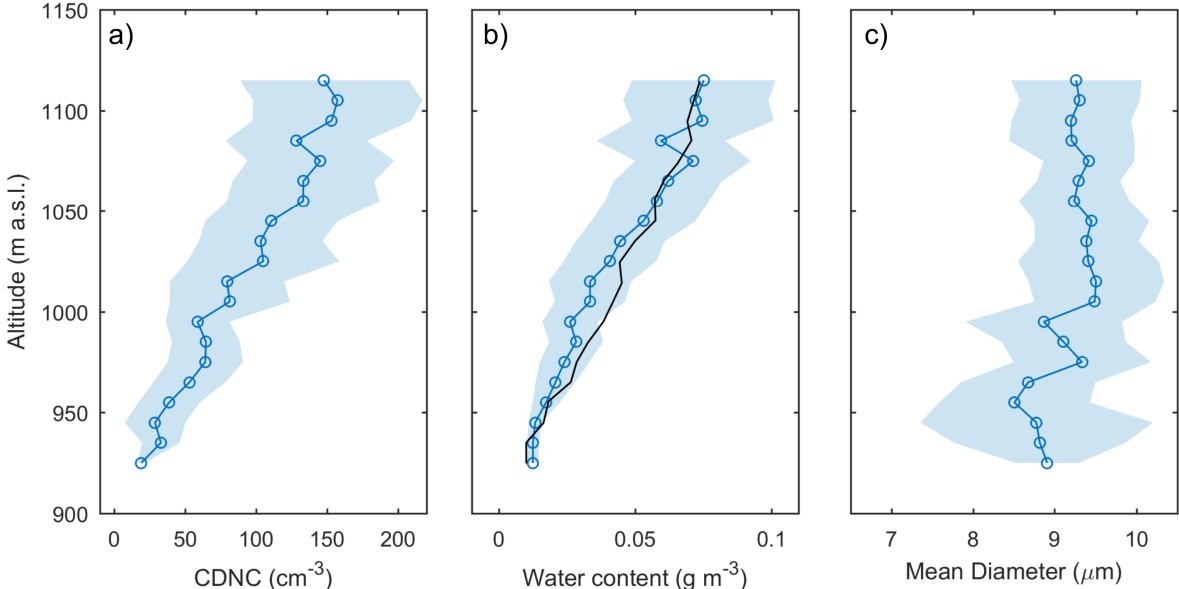

**Figure 7.** Mean vertical profiles of the cloud droplet number concentration (left), liquid water content (center) and mean cloud droplet diameter (right) averaged over the 9 profiles measured with the HoloBalloon platform. The data are averaged over an altitude interval of 10 m. The shaded area represents the standard deviation. The black line in b) shows the adiabatic LWC profile, which is calculated as follows: i) calculate the saturation vapor pressure $e_s$ at the cloud base, ii) use the pressure at the cloud base to determine the saturation mixing ratio $w_s(T,p) = \frac{\epsilon e_s(T)}{p - e_s}$, iii) calculate $w_t = w_s + w_l$ at the cloud base assuming $w_l$=0.01 and assuming constant $w_t$ with height (adiabatic), iv) calculate $w_s$ at all height levels, determine $w_l$ and multiply $w_l$ by local dry air density.

a positive Brunt-Väisälä frequency $N$ is observed (Fig. 6.e). These conditions are favorable for the development of boundary layer waves and Kelvin-Helmholtz instability (see Sect. 4.4). With a Brunt-Väisälä frequency $N$ of $0.04\,\mathrm{s}^{-1}$, the period of oscillation is $\tau = 2\pi/N = 2.5$ min, which lies in the range of typical wave periods of gravity waves in the boundary layer (few minutes up to one hour, Rees et al. 2000).

5  ## 4.3 Microphysical cloud structure

Figure 7 shows the mean vertical profiles of the microphysical cloud properties averaged over 9 profiles. The mean cloud droplet number concentration (CDNC) increases from $10\,\mathrm{cm}^{-3}$ at the cloud base (920 m) to $150\,\mathrm{cm}^{-3}$ at 1100 m (Fig. 7.a). The mean liquid water content (LWC) ranges between $0.01\,\mathrm{gm}^{-3}$ and $0.08\,\mathrm{gm}^{-3}$ and on average approaches an adiabatic profile (Fig. 7.b). The mean cloud droplet diameter decreases slightly from 9 μm to 8.5 μm in the first 30 m and stays constant

10  at around 9.5 μm above 1000 m (Fig. 7.c). The variability in the mean diameter is larger close to the cloud base. The observed CDNC of $150\,\mathrm{cm}^{-3}$, LWC of $0.08\,\mathrm{gm}^{-3}$ and mean cloud droplet diameter of 9.5 μm are in the range for fog and continental stratus clouds (Lohmann et al., 2016), but rather at the lower end of the range. Despite the supercooled conditions, only a few





ice crystals were observed ($< 0.1\,\mathrm{L}^{-1}$).

The increase in CDNC with increasing height is in contrast to the theory of an adiabatic cloud profile and might be explained by different factors. An adiabatic cloud model assumes that cloud droplets activate at the cloud base and grow in size with increasing altitude. Thus, CDNC is expected to remain constant with height after the maximum supersaturation is reached.

There are several possibilities why this theoretical criterion is not met for the case study presented here. Firstly, HOLIMO does not detect cloud droplets smaller than $6\,\mu\mathrm{m}$. This can lead to an underestimation in CDNC, especially at cloud base where the droplets are the smallest. Secondly, an adiabatic cloud model assumes a constant updraft, but fluctuations in the updraft speed or turbulence could generate supersaturated conditions and activate cloud droplets at higher altitudes than cloud base. Thirdly, the increase in CDNC with height could be driven by radiative cooling at the cloud top by producing either supersaturation

and/or instabilities and thus turbulences within the cloud layer. Moreover, it has to be considered that only the first 200 meters of the cloud were observed (up to 1150 m a.s.l), whereas the cloud top extends up to 1500 m a.s.l. On the other hand, a database of stratus clouds (Miles et al., 2000) showed that the CDNC in continental clouds was more variable with height than in marine clouds where CDNC was determined near cloud base. They suggested that different microphysical and dynamical processes occur in continental clouds, which could lead to a deviation of the theoretical assumption of a constant CDNC. Therefore, it is

unclear whether the observed increase in CDNC is a measurement artefact or a real feature of the observed cloud.

### 4.4 Inhomogeneities in the microphysical cloud properties of stratus clouds

We investigate cloud inhomogeneities by analyzing the height-temporal evolution of CDNC and of the anomaly of CDNC (Fig. 8). The CDNC anomaly $\mathrm{CDNC^a}$ is calculated by dividing CDNC observed in the height interval $h$ by the mean CDNC in that height interval averaged over the 9 profiles ($\mathrm{CDNC_h^a} = \mathrm{CDNC_h}/\overline{\mathrm{CDNC_h}}$). $\mathrm{CDNC_h^a}$ reveals areas of higher and lower CDNC

than the average concentration and gives indications of the CDNC variability within the cloud. As Korolev and Mazin (1993), we define areas with $\mathrm{CDNC_h^a} < 0.5$ as regions of decreased CDNC and areas with $\mathrm{CDNC_h^a} > 1.5$ as regions of increased CDNC. The height-temporal evolution of CDNC shows cloud inhomogeneities in CDNC on different scales (Fig. 8). For example, profile 7 shows regions of lower CDNC, whereas profile 9 shows regions of higher CDNC compared to the mean profile. CDNC at 1100 m in profile 9 ($200\,\mathrm{cm}^{-3}$) is more than a factor 3 higher than in profile 7 ($60\,\mathrm{cm}^{-3}$). Assuming a mean wind

speed of $10\,\mathrm{m\,s}^{-1}$, these translates to an inter-profile variability in CDNC on a scale of 30 km. From an intra-profile variability perspective, on a scale of tens of meters, all profiles show alternating regions of higher and lower CDNC. For example, CDNC in profile 7 increases by a factor of 1.7 in the 10 m height interval between 1100 m ($58\,\mathrm{cm}^{-3}$) and 1120 m ($100\,\mathrm{cm}^{-3}$). It is likely that the observed variations in CDNC exceed statistical variations and are the result of different cloud processes. Upon further analysis, we investigate various physical processes, which could explain these cloud inhomogeneities on different scales.

The variability in CDNC on a scale of several kilometers might be explained by internal gravity waves in the boundary layer. As discussed for example by Wanner and Furger (1990), strengthening or weakening of the Bise due to dynamic or orographic effects could induce oscillations within the cold air and the formation of boundary layer waves at the cloud top. The questions is whether these boundary layer waves can propagate through the cloud and explain the variability in CDNC. The presence of wind shear and a positive Brunt-Väisälä frequency ($N = 0.04\,\mathrm{s}^{-1}$) at the inversion (see Fig. 6.e) suggests that Kelvin-



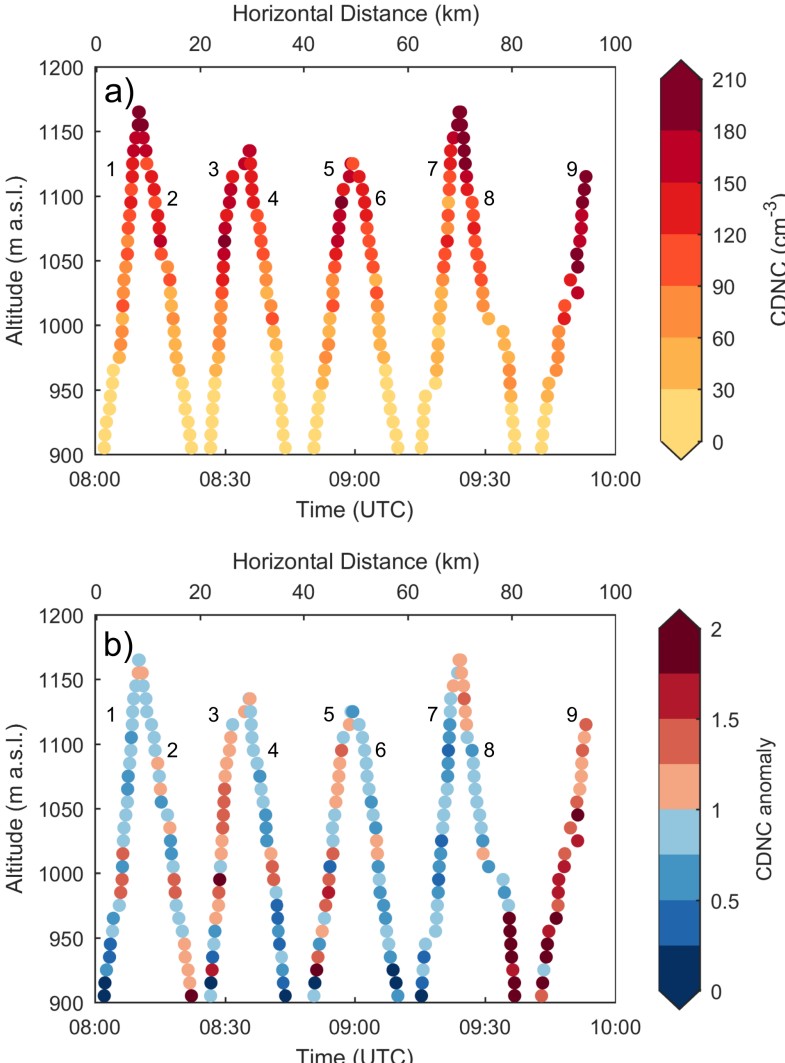

**Figure 8.** Height-temporal evolution of the CDNC (top) and the CDNC anomaly $CDNC_h^a$ (bottom) (see text for explanation of anomaly). The data points are averaged over an altitude interval of 10 m. The upper x-axis shows the horizontal distance $s$ of the cloud, assuming a mean wind speed $v$ of $10\,\mathrm{m\,s^{-1}}$ over time $t$ ($s = v \cdot t$). The numbers represent the profile number according to Table 2.

Helmholtz instability occurs at the cloud top, which is a favourable condition for the formation of boundary layer waves. These boundary layer waves can penetrate dry air into the cloud and could affect the cloud microphysics (e.g. Bergot 2013). The penetration depth depends on different factors such as the relative humidity above the cloud, the temperature stratification of the atmosphere, the turbulence conditions and the cloud water content (Korolev and Mazin, 1993). Kelvin-Helmholtz instability

5 can enhance mixing and modify entrainment at the cloud top (Mellado, 2017) and thus influence the cloud microstructure. However, microphysical observations up to cloud top and an extended set of wind measurements over a time period of several

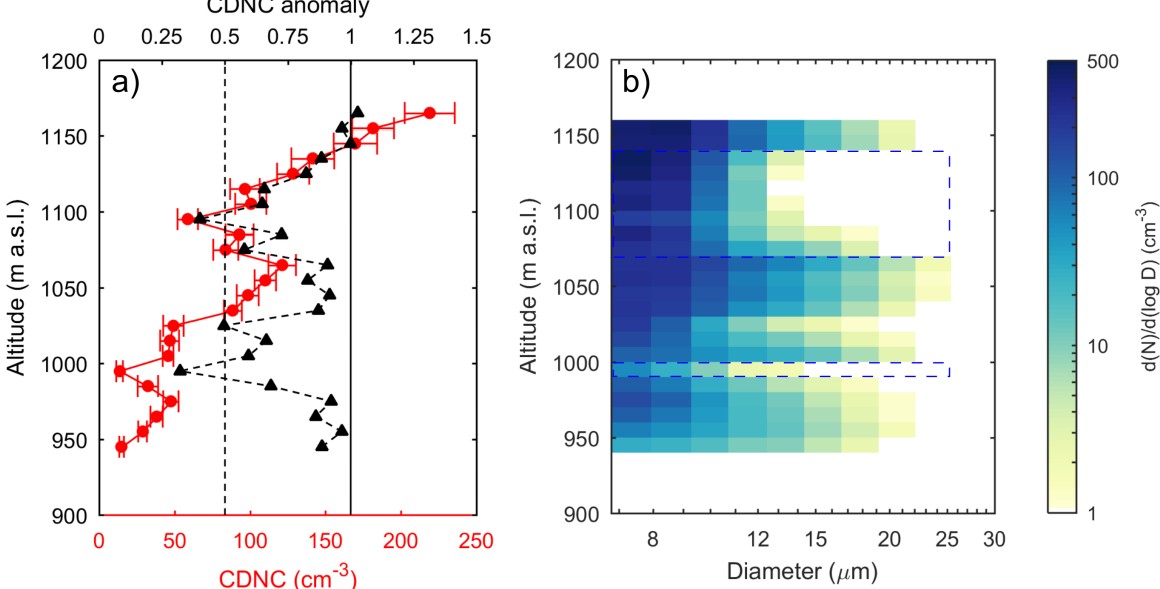

**Figure 9.** Vertical profile of CDNC and CDNC anomaly $CDNC_h^a$ (left) and number size distribution (right) of profile 7. The CDNC is shown by the red solid line and the CDNC anomaly by the black dashed line (see text for explanation of anomaly). The black dashed vertical line indicates regions of decreased CDNC ($CDNC_h^a < 0.5$). The data is averaged over a 10 m interval. The dashed rectangles in the number size distribution plot show regions of decreased CDNC (discussed in text).

hours would be necessary to further test the hypothesis of boundary layer waves influencing the structure of clouds.

Cloud inhomogeneities in CDNC on a meter scale can be the result of different processes, depending on their location within the cloud. We will discuss these cloud inhomogeneities based on the profile 7, because it shows regions of decreased CDNC (see Fig. 8.b). The cloud droplet number size distribution of profile 7 shows a gradual increase in cloud droplet size and number

concentration with height until a sudden decrease in particle concentration and size occurs between 1070 m and 1130 m (Fig. 9.b). In this region, CDNC is less than half of the average CDNC ($CDNC_h^a < 0.5$, Fig. 9.a). In addition, the cloud droplet spectrum shows an increase of CDNC of small droplets and an absence of cloud droplets larger than 14 µm. Korolev and Mazin (1993) propose several mechanisms for the formation of cloud inhomogeneities on a meter scale such as (i) entrainment, (ii) variability of the condensation level and (iii) evaporation in descending motions. Considering the location of our region of

decreased CDNC (300-400 m from cloud top, 200 m from cloud base), we assume that is most likely formed by evaporation in descending motions. The temperature inside a descending air parcel increases due to adiabatic compression and heating and in response cloud droplets evaporate leading to regions of decreased CDNC. According to the calculations in Korolev and Mazin (1993), a vertical displacement $\Delta Z^*$ of 60 m would be necessary for complete droplet evaporation of a cloud parcel with an initial LWC of 0.1 g kg$^{-1}$, assuming no mixing and an adiabatic gradient of specific LWC $\beta_{ad}$ of $1.6 \cdot 10^{-3}$ g kg$^{-1}$

(T = 273 K and p = 900 mb). As we observed similar initial conditions, these calculations support the hypothesis that the region





of decreased CDNC between 1070 m and 1130 m was formed by droplet evaporation in a descending air parcel. Even though radiative cooling or the Kelvin-Helmholtz instability could enhance entrainment at the cloud top, it is unlikely that entrainment influences the cloud structure 300-400 m below the cloud base.

Similarly, a sudden decrease in the CDNC is observed at around 1000 m. Unlike the decrease between 1070 m and 1130 m, a decrease of both the small and large droplets is observed, suggesting that another mechanism is occurring. Based on the location within the cloud and the decrease of small and large droplets, we assume that this region of decreased CDNC could be explained by irregularities of the condensation level (e.g. Korolev and Mazin 1993). These irregularities could be due to fluctuations in temperature or humidity close to the cloud base. This could lead to homogeneous mixing of the cloudy and cloud-free volumes and thus to a decrease of both CDNC and size.

## 5   Discussion

### 5.1   Clouds - a complex multi-scale phenomenon

Even though high fog and stratus clouds are of stratiform nature, we found that stratus clouds are complex dynamic structures with cloud inhomogeneities on different scales. The results of the case study are summarized and conceptualized in Fig. 10. These cloud inhomogeneities represent and influence various processes on a wide range of scales.

Inhomogeneities on scales of several kilometers can play an important role for cloud radiative effects (Slingo, 1990). These inhomogeneities can be formed for example by internal gravity waves in the boundary layer due to topographic effects or dynamical instabilities. We found conditions favorable for the formation of boundary layer waves and Kelvin-Helmholtz insta-bilities at the cloud top (e.g. wind shear, stable stratification). The Brunt-Väisälä frequency indicates a wave period of around 2-3 minutes, which is at the lower range of typical boundary layer waves. These boundary layer waves could propagate into the cloud (Fig. 10, bottom left), influence the cloud microphysical processes and explain the observed variabilities in CDNC on a scale of tens of kilometers (Fig. 10, top left). Previous studies suggested that boundary layer waves and Kelvin-Helmholtz in-stabilities near the cloud top could affect the cloud microphysics. For example, a large-eddy simulation study by Bergot (2013) showed that eddies near the cloud top influenced the fog microphysics of the fog layer and that the observed fluctuations in LWC close to the cloud top can have a strong impact on the radiative fluxes.

Cloud inhomogeneities on a meter scale can be the result of different processes. For example, we observed a sudden decrease in CDNC and cloud droplet size in a 50 m height interval within the cloud. We assume that this is the result of adiabatic compres-sion and heating and subsequent droplet evaporation in a downdraft region of an eddy (Fig. 10, bottom center). Furthermore, we observed a region of decreased CDNC and size close to the cloud base. We hypothesize that this region of decreased CDNC is formed by irregularities in the condensation level. Previous studies have found cloud inhomogeneities in cloud properties on a meter scale (e.g. Korolev and Mazin 1993, Gerber et al. 2005). For example, Gerber et al. (2005) found regions with sharply reduced LWC compared to the background in stratocumulus clouds ('cloud holes'), which were assumed to be the result of entrainment of dry air. Similarly, García-García et al. (2002) observed regions of decreased CDNC of a few tens of meters in warm fog characterized by a broader droplet size distribution. These inhomogeneities on a meter scale can influence the cloud

**Figure 10.** Overview image summarizing the observed cloud inhomogeneities on different scale (top) and discussing multiple cloud processes, which could explain these cloud inhomogeneities (bottom).





droplet size distribution and thus play an important role in the evolution of the cloud.

Inhomogeneities on the centimeter and millimeter scale can influence precipitation initiation and the radiative properties of clouds. Several studies (e.g. Baker 1992, Brenguier 1993, Beals et al. 2015) observed sharp transitions at the interface between cloud and ambient air on the centimeter scale as a response to entrainment and mixing. Moreover, these inhomogeneities on the

microscale can influence growth by collision-coalescence or the Wegener-Bergeron-Findeisen process and thus the efficiency of precipitation formation. The ability of HOLIMO to detect the spatial distribution of particles in a cloud volume allows studying these small-scale processes and particle-particle interactions on a millimeter scale (Fig. 10, top right). However, a quantitative analysis of the spatial distribution (e.g. Larsen and Shaw (2018), Larsen et al. (2018)) is required to assess these small-scale processes, which is beyond the scope of this study.

The results presented here and previous studies show that cloud inhomogeneities on different scales occur in clouds. It is still not fully understood how these cloud inhomogeneities are formed, how these inhomogeneities influence the evolution of the cloud structure and how they interact on different scales. With HoloBalloon, we bring together a wide range of scales from the kilometer down to the millimeter scale. Such a multi-scale approach could help to improve the understanding the inhomogeneous, dynamic and complex nature of clouds.

## 5.2 Deploying a TBS in the field

The HoloBalloon platform was successfully deployed in various meteorological conditions. In situ profiles up to 700 m altitude above the ground were obtained, limited by air traffic restrictions in the maximum altitude. Unfortunately, because of this limitation in the maximum altitude, we were not able to penetrate the whole cloud layer and perform measurements at the cloud top. The platform was deployed at temperatures down to -8 °C. Despite the supercooled conditions, we observed only a few ice

crystals ($< 0.1\,\mathrm{L}^{-1}$). Even though parts of the balloon and of the cable were covered in ice, this did not affect our measurements and the flight performance. However, based on our experience, we recommend covering the balloon with a tarpaulin during night to prevent accumulation of snow and water on the balloon. We flew the TBS in wind speeds up to $15\,\mathrm{m\,s}^{-1}$. The TBS was stable in these high wind conditions, but the ground handling became challenging at wind speeds above $10\,\mathrm{m\,s}^{-1}$, especially in the presence of wind gusts.

Regarding the effort for setting up and operating the HoloBalloon platform, several aspects should be considered. Firstly, a closed air space was required to perform cloud measurements with a TBS. The process of obtaining a closed air space was closely coordinated with the aviation safety authority. In areas with dense air traffic, such as the Swiss Plateau, it can be difficult to find a suitable location. Secondly, a large, reasonably flat surface area ($\sim20\times40\,\mathrm{m}$) is required to prepare and launch the TBS. No major obstacles (e.g. trees, power lines) should be within a radius of around 60 m of the launching site and it should

be possible to insert an anchor into the ground. The system set up takes approximately 3 days and requires 2-3 trained persons for operation. A third person can especially be helpful during difficult wind conditions.

HoloBalloon was able to measure temperature, relative humidity and wind profiles in boundary layer clouds. In general, the measurements agreed well with the observations from the MeteoSwiss weather stations and the radiosondes (see Fig. 6). The temperature sensor showed a delayed response to changes in the ambient temperature (not shown), as was observed on the



cable car platform HoloGondel (Beck et al., 2017). To overcome this issue, the temperature was calculated from the virtual temperature of the 3D sonic anemometer, assuming water saturation in the cloud. It is well known that it is difficult to measure relative humidity in clouds (e.g. Korolev and Mazin 2003, Korolev and Isaac 2006). The relative humidity measured by the HoloBalloon platform in clouds ranged between 93 % and 98 %. We assumed in-cloud conditions when the relative humidity remained constant with height. Wind speed and direction measurements were corrected for the motion of the balloon. As described in Sect. 2, this was done using the output from an inertial navigation system and a GPS antenna following the procedure described in Mellado (2017). The corrected horizontal wind speed and wind direction measurements agreed well with the radiosonde observations. The vertical wind speed was not considered in this study. Although turbulence was not the focus of this study, turbulence measurements should be interpreted with caution. As the instrument package was installed on the kiel below the balloon (Fig. 1), we cannot exclude an influence from the balloon. For future field campaigns, the feasibility of installing the instrument package 30-40 m below the balloon should be assessed.

The vertical profiles of the microphysical measurement showed no systematic difference between ascending and descending profiles (see Fig. 8.b), suggesting that the balloon was not significantly influencing the microphysical measurements. With a mean horizontal wind speed of $10\,\mathrm{m\,s^{-1}}$ and a cable speed of $1\,\mathrm{m\,s^{-1}}$, the horizontal wind speed is by a factor 10 larger than the cable speed. This, in combination with a flight angle of up to 45° (due to the kytoon design), prevents shading effects and further support the assumption that a 'pristine' cloud volume is measured.

Generally, the measured size distributions showed the maximum number concentration close to the resolution limit of HOLIMO. This demonstrates the limits of the instrument in measuring small cloud particles ($<6\,\mu\mathrm{m}$). This bias can lead to an underestimation of CDNC, especially close to cloud base. For future field campaigns, instruments measuring cloud particles below $6\,\mu\mathrm{m}$ would be helpful (e.g. optical particle counter), especially in fog or clouds with small mean cloud diameters in order to cover the entire cloud droplet spectrum.

## 6 Conclusions

HoloBalloon has proven its ability to measure in situ vertical profiles of microphysical and meteorological cloud properties of boundary layer clouds up to 700 meters above ground. Here, we presented observations of a supercooled low stratus cloud during a Bise event over the Swiss Plateau in February 2018. Our main findings are summarized as follows:

– HoloBalloon merges the advantages of holography with the benefits of a TBS. Unlike other single cloud particle instrumentation, holographic cloud imagers have a well-defined sample volume independent of particle size and air speed despite fluctuating aspiration speeds on a TBS. The low aspiration speed on the TBS in combination with the high acquisition rate of HOLIMO 3B allows for measurements with high spatial resolution.

– The HoloBalloon platform was successfully deployed at temperatures down to -8 °C and wind speeds up to $15\,\mathrm{m\,s^{-1}}$. While conventional blimp-like TBS are limited to wind speeds below $10\,\mathrm{m\,s^{-1}}$, kytoons are designed for wind speeds up to $25\,\mathrm{m\,s^{-1}}$, making them an interesting measurement platform for atmospheric research.



- HoloBalloon was able to reliable measure vertical profiles of the microphysical cloud properties and meteorological parameters. The meteorological measurements agreed well with observations from radiosondes and weather stations and the observed cloud properties were within the expected range. Cloud particles between 6 μm and 24 μm and CDNC up to 200 cm$^{-3}$ were observed with HOLIMO.

- HoloBalloon was able to capture cloud inhomogeneities on different scales. For example, we observed variability of CDNC from a kilometer down to a meter scale. Boundary layer waves or Kelvin-Helmholtz instability at the cloud top connected to the Bise situation influenced the cloud structure on a kilometer scale. Moreover, we observed cloud regions with decreased CDNC and cloud droplet size on a scale of 30-50 m, likely caused by droplet evaporation in a descending air parcel. In addition, HOLIMO is capable of measuring the spatial distribution of cloud particles in a cloud volume

on a millimeter scale (e.g. Beals et al. 2015, Beck et al. 2017). This outstanding feature of holography allows studying processes on the particle scale such as mixing of cloudy and dry air or growth by collision-coalescence, which can be important for precipitation formation.

Lastly, the modular design of the HoloBalloon platform allows for varying research questions to be addressed. For example, the instrument package can be modified to include aerosol, radiation or turbulence measurements. Additionally, the measurement

platform could be installed 30-50 m below the balloon to avoid any interference from the balloon itself. Furthermore, future field campaigns will include an optical particle counter to extend the cloud particle spectrum to particles below 6 μm.

*Code and data availability.*

*Author contributions.* FR prepared the manuscript with contributions from AB, JH and UL. FR and JH performed the HoloBalloon measurements. FR analyzed the HoloBalloon measurements and FR, AB, JH and UL interpreted the data.

*Competing interests.* The authors declare that they have no conflict of interest.

*Acknowledgements.* The authors would like to thank Jörg Wieder for his assistance during the size calibration experiments and Julie Pasquier for the analysis of the resolution measurement. We also thank Hannes Wydler and Michael Rösch for their technical support in designing the HoloBalloon platform. The authors also thank the Kieswerk Risi and the Gemeinde Aarwangen for their excellent support during the field campaign. We would also like to thank the Federal Office of Civil Aviation (FOCA) for their assistance in getting the flight permit. The

meteorological measurements were provided by the Swiss Federal Office of Meteorology and Climatology MeteoSwiss. This project was supported by the ETH Scientific Equipment Program.



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
