# Peer review of "Using a holographic imager on a tethered balloon system for microphysical observations of boundary layer clouds"

_Atmospheric Measurement Techniques, 2019_

## Referee Comment (RC1) · Anonymous Referee #1 · 19 Sep 2019

**1   Main review points**

- In this manuscript, the authors present a holographic imaging system and its application to the analysis of low stratus properties in a case study from Switzerland. The paper is well-written, has a clear structure, and overall presents a good overview of the potential of the technique in studying boundary-layer clouds. The case study presented includes some very interesting aspects, the resulting hypotheses are summarized in a useful conceptual sketch. Any generalization would require further samples, but is beyond the scope of this paper focused on the introduction of the technique.

[Figure]

- Language: The paper is legible and understandable, but riddled with small lingual errors that could probably be corrected quickly by someone fully proficient in English.

**2 Details**

- page 1, line 3 (1-3): which cloud properties exactly?

- 1-4: since holographic imagers are not a common type of instrument in large parts of the cloud community, please add a very short note on the principle in the abstract

- 1-10: scales have been mentioned in line 7 alrady, but in contradiction to this line

- 1-11: I think an example is not needed in the abstract

- 2-10: What do you mean by "most of the observations", and how did you reach this conclusion?

- 2-13: is there a source for this (problems in lowermost km)?

- 2-21: what is "ice shattering", and how does it impact measurements?

- 3-3: some aditional info on the principles of holography would be useful here

- 3-18: i.e. a low stratus cloud with its cloud base above ground? Please specify.

- 3-20: how do you define inhomogeneity here?

- 5-27: What do you use as training data?

- 5-28: How are these parameters calculated?

- 5-28: "such as" – please be specific here and list all parameters.

- 9-5: Why was this particular situation chosen? In what ways is it representative or not?

- Figure 5: Please provide complete citation (author, year)

- Table 2: Why is there no descent for profile number 9?

- 11-18: What do you mean by classification in this context? Which classes?

- 11-18: How is a classification by hand performed?

- Section 4.3: How do you explain the nearly constant with height droplet diameters?

- 17-12: I think this statement is too general, given that only one case is analyzed.

**3 Technicalities**

- page 1, line 1 (henceforth 1-1 etc.): aircrafts → aircraft

- 1-2: orographically diverse

- 1-2: densely populated

- 1-5: velocity-independent sample

- 1-6: allows for observations

- 1-7: scales

- 1-9: above the ground were performed at temperatures...

- 1-11: scales (No more comments on language from this point forward)

---

## Referee Comment (RC2) · Anonymous Referee #2 · 17 Dec 2019

The paper describes balloon-borne measurements of microphysics inside supercooled boundary layer stratus clouds collected with use of a modern holographic imager HOLIMO.

The paper consists of two very distinct parts. From the beginning to section 4.3 the paper is clear, very well written and there are no major drawbacks in the text. The description of the measurements, calibration is sound. The results are interesting, show unexpected behavior of cloud microphysics, hard to document with different, than HOLIMO, instruments. This, with some additional discussion and maybe selected examples of local samples of droplet spatial and size distributions would be enough to justify the

publication. However, instead of focusing on microphysics, in the last sections of chapter 4 and in the discussion author speculate on mixing and dynamical effects which are aimed at explanation the unexpected results of microphysical measurements, in particular large variations in droplet number concentration. These speculations should be backed with the data, but are not. As shown in Fig.1 the HoloBallon is, together with the HOLIMO, equipped with a sonic anemometer, which should provide in-situ high-resolution data on turbulence (velocity fluctuations) and virtual temperature. The authors, instead of using data from the device, speculate on turbulence and waves, Kelvin-Helmholz Instabilities, downdrafts. I strongly believe that insight into sonic data could be used to verify which speculations are justified and which are not. In particular virtual temperature fluctuations might help to understand mixing, velocity records should allow to document turbulence, waves and K-H instabilities

In my opinion the paper in the present form is hardly acceptable. I suggest the major revision of the text. Two options is possible: 1) to make the paper shorter, remove the speculative part of the chapters 4 and 5 and to write that the explanation requires additional, highly demanding analysis of turbulence data recorded; 2) to use sonic data and do the analysis in a simplified form, to show some dynamical properties of the flow to support speculations presented in the text.

If the authors chose the second option I suggest more detailed insight into the cited Mellado's paper about stratocumulus top and into references therein. Such insight, in my opinion, could help very much in the analysis.
* * *

---

## Author Comment (AC1) · 14 Jan 2020

Reviewer comments on '**Using a holographic imager on a tethered balloon system for microphysical observations of boundary layer clouds**' by Fabiola Ramelli, Alexander Beck, Jan Henneberger, Ulrike Lohmann
*Response to Reviewer #1*

We would like to thank the anonymous referee for his/her valuable feedback and suggestions on the paper. We incorporated the suggestions within the revised manuscript, which substantially improved the quality of the manuscript. In the following, we will address the comments and show the changes in the revised manuscript.

**General comments**

1) In this manuscript, the authors present a holographic imaging system and its application to the analysis of low stratus properties in a case study from Switzerland. The paper is well-written, has a clear structure, and overall presents a good overview of the potential of the technique in studying boundary-layer clouds. The case study presented includes some very interesting aspects, the resulting hypotheses are summarized in a useful conceptual sketch. Any generalization would require further samples, but is beyond the scope of this paper focused on the introduction of the technique.

2) Language: The paper is legible and understandable, but riddled with small lingual errors that could probably be corrected quickly by someone fully proficient in English

**Detailed comments**

1) **1-3: which cloud properties exactly?**
   Thank you for pointing this out. We have added the cloud properties as follows (page 1, line 6-7): "*Based on a set of two-dimensional images, information about the phase-resolved particle size distribution, shape and spatial distribution can be obtained.*" More information about the cloud properties are provided in Sect. 1 and Sect. 3.1.

2) **1-4: since holographic imagers are not a common type of instrument in large parts of the cloud community, please add a very short note on the principle in the abstract**
   Thank you for pointing this out. We agree that holographic imagers are not a common type of instrument in large parts of the cloud community. We added a short description of the working principle in the abstract (page 1, line 4-8). Additionally, more information about the working principle of in-line holography can be found in Section 3.1.

3) **1-10: scales have been mentioned in line 7 already, but in contradiction to this line**
   Thank you for pointing this out. In the following study, only measurements down to the meter scale are presented. However, holographic imagers can provide information down to the millimeter scales if the spatial distribution of cloud particles is analyzed. The spatial

distribution of particles is not analyzed in the presented case study, but its potential is highlighted in Sect. 5.2.

4) **1-11: I think an example is not needed in the abstract**
Thank you for the comment. The examples have been removed in the revised manuscript.

5) **2-10: What do you mean by "most of the observations", and how did you reach this conclusion?**
Thank you for the comment. The term 'most' might be inadequate. We changed it accordingly to *'a large fraction of the observations'* (page 2, line 9). Moreover, we included some references, which used satellite observations to study boundary layer clouds.

6) **2-13: is there a source for this (problems in lowermost km)?**
Thank you for pointing this out. We included two references describing the problem of surface clutter (page 2, line 14).

7) **2-21: what is "ice shattering", and how does it impact measurements?**

Thank you for the comment. We included a short description of ice shattering and how it can impact the measurements (page 2, line 23-24): *"Ice shattering occurs if an ice crystal impacts the instrument tips or an inlet prior to entering the detection volume, which can result in a large number of small ice particles being a measurement artefact."*

8) **3-3: some additional info on the principles of holography would be useful here**
Thank you for the comment. We add a reference to Section 3 (page 3, line 11), where a more detailed description of the holographic instrument and holography is provided.

9) **3-18: i.e. a low stratus cloud with its cloud base above ground? Please specify.**
Thank you for pointing this out. We removed the term high fog throughout the whole paper and replaced it by stratus clouds, which is the more general term. The term 'high fog' is mainly used in Switzerland and therefore introduced at the beginning of the case study (page 10, line 2).

10) **3-20: how do you define inhomogeneity here?**

Thank you for the comment. We added a definition of inhomogeneity (page 3, line 30-31): *"Throughout this study, inhomogeneities are defined by the variability in the cloud droplet number concentration and cloud droplet size."*

11) **5-27: What do you use as training data?**
12) **5-28: How are these parameters calculated?**
13) **5-28: "such as" – please be specific here and list all parameters.**

Thank you for the comments. The comments 11-13 are addressed together. In order to provide more details about the particle classification, we extended the description as follows (page 5, line 27-33): "*The resulting 2D shadowgraphs can be classified as cloud droplets, ice crystals and artefacts based on a set of parameters using supervised machine learning (e.g. Fugal et al. 2009, Beck et al. 2017, Touloupas et al. 2019). In the present study, a set of 6400 particles was classified manually, which served as a training data set on support vector machines. From the classification, the phase-resolved particle size distribution can be computed. The particle diameter is calculated based on the number of pixels (see also Sect. 3.3) and the number concentration can be computed from the particle counts within the well-defined sample volume. Only particles that exceed a size of 2x2 pixels (6 µm) are considered.*"

14) **9-5: Why was this particular situation chosen? In what ways is it representative or not?**

Thank you for the comment. The presented stratus cloud event is representative for a Bise situation, which often occurs during winter (page 10, line 3-4). Below, we added a figure from Wanner and Furger (1990), which summarizes the frequency of wind direction from radiosonde ascents launched from Payerne for the period 1981-1985. Based on their result, Bise occurred on 27% of the hours (see also Weber and Furger, 2001 or MeteoSwiss).

Wanner, H., & Furger, M. (1990). The bise—climatology of a regional wind north of the Alps. *Meteorology and Atmospheric Physics*, 43(1-4), 105-115.
Weber, R. O., & Furger, M. (2001). Climatology of near-surface wind patterns over Switzerland. *International Journal of Climatology: A Journal of the Royal Meteorological Society*, *21*(7), 809-827.
MeteoSwiss:https://www.meteoschweiz.admin.ch/content/dam/meteoswiss/de/service-und-publikationen/Publikationen/doc/Web_Wetterlagen_DE_low.pdf

[Figure]

Fig. 4. Frequency of wind directions for 150 m layers from 500 to 6 200 m ASL for radiosonde ascents from Payerne for the period 1981–1985

**15) Figure 5: Please provide complete citation (author, year)**

Thank you for the comment. We added the year and the link where the online maps can be downloaded (https://www.atlasderschweiz.ch/) (caption Fig. 5).

**16) Table 2: Why is there no descent for profile number 9?**

Thank you for the comment. There is no profile 10/ no descent for profile number 9, because the battery of the instrument package was empty. We added a sentence to specify that (page 11, line 18-19): *"The battery of the instrument package was empty after profile 9, thus no observations were available afterwards."*

**17) 11-18: What do you mean by classification in this context? Which classes?**

**18) 11-18: How is a classification by hand performed?**

Thank you for the comments. The particles are classified into three classes (cloud droplets, ice crystals and artefacts). We added the classes in the text and added a reference to Sect. 3.1 (page 12, line 4), where the classification process is described in more detail. Moreover, we exchanged the term 'classification by hand' with the term *'classified manually (visual classification)'* (page 12, line 5).

**19) Section 4.3: How do you explain the nearly constant with height droplet diameters?**

Thank you for pointing this out. Figure 7.c) shows the mean vertical profile of the cloud droplet diameter. We agree that in the mean, the cloud droplet diameter looks rather constant. Here we include a figure that shows the individual vertical profiles of the cloud droplet diameter. It can be seen that in general the mean cloud droplet diameter increases with height. Profile 7 shows a lower mean diameter than the other profiles at altitudes above 1050 m.

Furthermore, it is also possible that there is a higher competition for water vapor with increasing height due to the increasing CDNC.

[Figure]

**20) 17-12: I think this statement is too general, given that only one case is analyzed.**

Thank you for the comment. We agree that this statement is too general, since the analysis is based on the observations of only one case study. We adapted the sentence in the following way (page 18, line 19-20): "*We found that stratus clouds can exhibit complex dynamic structures with microphysical signatures on different scales (Sect. 4.4).*"

**Technicalities**

Thank you for all the technical comments

1) page 1, line 1 (henceforth 1-1 etc.): aircrafts→aircraft
   Changed to aircraft (page 1, line 1)
2) 1-2: orographically diverse
   Changed (page 1, line 2)
3) 1-2: densely populated
   Changed (page 1, line 2)
4) 1-5: velocity-independent sample
   Changed (page 1, line 7)
5) 1-6: allows for observations
   Changed (page 1, line 9)
6) 1-7: scales
   We think that 'scale' should be used in singular in this case.
7) 1-9: above the ground were performed at temperatures...
   Changed (page 1, line 13)
8) 1-11: scales (No more comments on language from this point forward)
   We think that 'scale' should be used in singular in this case.

---

## Author Comment (AC2) · 14 Jan 2020

Reviewer comments on '**Using a holographic imager on a tethered balloon system for microphysical observations of boundary layer clouds**' by Fabiola Ramelli, Alexander Beck, Jan Henneberger, Ulrike Lohmann
*Response to Reviewer #2*

We would like to thank the anonymous referee for his/her valuable feedback and suggestions on the manuscript. We incorporated the suggestions within the revised manuscript, which substantially improved the quality of the manuscript. In the following, we will address the comments.

**General comments**

*The paper describes balloon-borne measurements of microphysics inside supercooled boundary layer stratus clouds collected with use of a modern holographic imager HOLIMO. The paper consists of two very distinct parts. From the beginning to section 4.3 the paper is clear, very well written and there are no major drawbacks in the text. The description of the measurements, calibration is sound. The results are interesting, show unexpected behavior of cloud microphysics, hard to document with different, than HOLIMO instruments. This, with some additional discussion and maybe selected examples of local samples of droplet spatial and size distributions would be enough to justify the publication. However, instead of focusing on microphysics, in the last sections of chapter 4 and in the discussion author speculate on mixing and dynamical effects which are aimed at explanation the unexpected results of microphysical measurements, in particular large variations in droplet number concentration. These speculations should be backed with the data, but are not. As shown in Fig.1 the HoloBallon is, together with the HOLIMO, equipped with a sonic anemometer, which should provide in-situ high-resolution data on turbulence (velocity fluctuations) and virtual temperature. The authors, instead of using data from the device, speculate on turbulence and waves, Kelvin-Helmholz Instabilities, downdrafts. I strongly believe that insight into sonic data could be used to verify which speculations are justified and which are not. In particular virtual temperature fluctuations might help to understand mixing, velocity records should allow to document turbulence, waves and K-H instabilities.*

*In my opinion the paper in the present form is hardly acceptable. I suggest the major revision of the text. Two options is possible: 1) to make the paper shorter, remove the speculative part of the chapters 4 and 5 and to write that the explanation requires additional, highly demanding analysis of turbulence data recorded; 2) to use sonic data and do the analysis in a simplified form, to show some dynamical properties of the flow to support speculations presented in the text.*

*If the authors chose the second option, I suggest more detailed insight into the cited Mellado's paper about stratocumulus top and into references therein. Such insight, in my opinion, could help very much in the analysis.*

Thank you very much for your valuable comments and suggestions. In the substantially revised manuscript (especially Sect. 4.4 and Sect. 5), we focus more on the technical aspects of the HoloBalloon platform, rather than on the scientific outcome of Sect. 4.4. We agree that further data and analysis are required to back the hypotheses presented in Sect. 4.4. As the aim of the paper is (1) to introduce and characterize the newly developed HoloBalloon platform, (2) to provide a proof of concept for the HoloBalloon platform (case study) and (3) to show the potential and limitations of the platform in studying boundary layer clouds, we shortened the section 4.4 (especially the speculative part) as well as the discussion of it. Furthermore, we clearly indicate that further analysis of our data as well as auxiliary data (e.g. three-dimensional wind field, turbulence) are required to study the proposed mechanisms/ test the hypotheses, which lies beyond the scope of this paper.

We agree that the data of the 3D sonic anemometer could provide useful information about the dynamical properties of the flow and help to support the proposed hypotheses. However, we decided to not analyze the turbulence data of the 3D sonic anemometer, as it is installed on the keel below the balloon. Several experts in the field advised us to install the instrument package in future 20-30 m below the balloon in order to reduce influences of the balloon on the turbulence measurements. Thus, in the present study we cannot exclude influences of the balloon on the turbulence measurements (described in Sect. 5.1). For future field campaigns, we will follow the advices and install the instrument package 20-30 m below the balloon to be able to analyze turbulence data of the 3D sonic anemometer. The feasibility of a hanging mount was already successfully tested in the field in autumn 2019.

In the revised manuscript, we changed the order of subsections 5.1 and 5.2 in the discussion section. Moreover, we completely rewrote Sect. 5.2 (previous 5.1). In Sect. 5.2, we focus more on the technical aspect of HoloBalloon, rather than on the scientific outcome of Sect. 4.4 (in contrast to the previous version). The observations of the presented case study are used as an example to discuss the potentials and limitations of the HoloBalloon in studying boundary layer clouds.